# Determinants of the use of modern contraceptives among women of reproductive age group in Ethiopia: A multi-level mixed effects analysis

**Molalign Gualu Gobena** **, Maru Zewdu Kassie**

Statistics, Assosa University, Assosa, Benishangul Gumez Reginal State, Addis Ababa, Ethiopia

* molaligngualu@gmail.com

## Abstract

### Introduction

Modern contraceptive methods are a scientifically effective method to control the fertility of reproductive-aged groups of people. The women's use of contraceptive methods creates a birth gap and limits the number of their children. The main objective of this study is to identify the significant determinant of modern contraceptive use of reproductive-aged women in Ethiopia.

### Methods

We used data from 2019 Ethiopian Mini Demographic and Health Survey. This data was multi-level, taking into account factors at the individual and community levels. In order to capture the multi-level structure of this data and make more reliable and broadly applicable conclusions about the variables influencing the use of modern contraceptives at the individual and community levels, we employed a two-level mixed-effects logistic regression model. In addition, we used cross-tabulation analysis to know the percentage of modern contraception users (reproductive-aged women) across their socio-economic, demographic, and health characteristics. A total of 8196 reproductive aged (15–49) women were included in this study.

### Results

From a total of 8196 reproductive-aged women, 2495(30.4%) were using modern contraceptive method and the rest 5701(69.6%) did not use any modern contraceptive methods. Among 2495 contraceptive users, 1657 (67.3%) used injections and 533 (21.7%) used implants/Norplant. At a 5% level of significance, the result from the two-level binary logistic regression model revealed that the predictors; Age of women, education level, religion, wealth index, knowledge of modern contraception method, number of died children, number of living children, family size, total children ever born and contextual region have significant effect on the use of modern contraception method.

**Data Availability Statement:** Free registration and explanation for the use of data are required to access the data freely at https://dhsprogram.com/

data/dataset/Ethiopia_Interim-DHS_2019.cfm?
flag=0.

**Funding:** The author(s) received no specific funding for this work.

**Competing interests:** The authors have declared that no competing interests exist.

**Abbreviations:** EMDHS, Ethiopian Mini Demographic and Health Survey; MSAP, multicomponent sexual activity profile; EAs, Enumeration Areas; Df, Degree of Freedom; AIC, Akaike information criterion; BIC, Bayesian information criterion.

## Conclusion

Reproductive-aged women in Ethiopia with more living children, residing in urban/agrarian region, younger, wealthier, married, and more educated, were more likely to be modern contraceptive users. The concerned bodies in Ethiopia should bring forward the intervention strategy and should expand the existed programs to improve the use of modern contraception methods among reproductive-aged women in Ethiopia. Especially, they should give special attention to reproductive-aged women of less income, resident in pastoralist region, less educated, unmarried, and haven't living child.

## Introduction

Modern contraceptive methods are a scientifically effective method to control the fertility of reproductive-aged groups of people [1–3]. The women's use of contraceptive methods creates a birth gap and limits the number of their children [2,4]. Consequently, they would have better health with their children [1]. Implants, female and male condoms, injectables, contraceptive pills, standard days method, male and female sterilization, intrauterine devices, and emergency contraception are some of the modern methods of contraception [1,3]. The use of contraceptives had aims beyond merely avoiding unintended pregnancies. For instance, it aimed to: promote individual and family well-being; advance gender equality; support poverty reduction; contribute to sustainable development; improve maternal and child health; and empower individuals to achieve their desired family size [1,2,5,6].

In 2019, 45% of reproductive-aged women around the globe used modern contraception methods. It accounts for 91% of all types of contraceptive users [3]. Most women in countries where contraception use is greater than 50% use modern contraception methods, while those in countries where contraception use is less than 50% use traditional contraception methods. Among low-level modern contraception user countries (less than 50%), the use of such methods for women was varied, and rely mainly on a traditional method of contraception to control unwanted pregnancy and to extend the birth interval [3,7].

Women's contraceptive use is less than 50% in most countries in Sub−Saharan Africa, Central & Southern Asia, Northern Africa & Western Asia, Oceania excluding Australia & New Zealand, and Eastern & South−Eastern Asia. In contrast, women's contraceptive use is greater than 50% in most countries in Europe & Northern America, Latin America & the Caribbean, and Australia & New Zealand [3,7,8]. In addition, there is a variation in using modern contraception methods for women in each region. For instance, among all women in Latin America & the Caribbean, the use of modern contraception is varied from 25% in Haiti to 68% in Cuba. In the same way, modern contraception usage for women in Sub-Saharan Africa lies between 4% in South Sudan and 52% in Eswatini [3].

Globally, women's use of modern contraceptive methods has increased. In recent years, this increment was significant in sub-Saharan Africa [2,7]. To imply numerically, 11 countries in sub-Saharan Africa show the greatest increment in modern contraception usage between 2010 and 2019. These countries with corresponding percentage increment of modern contraception usage from 2010 to 2019 in the bracket include Malawi (14.6%), Ethiopia (13%), Lesotho (12.3%), Kenya (12%), Sierra Leone (11.7%), Liberia (11.5%), Burkina Faso (11.1%), Senegal (10.6), Uganda (10.2), Madagascar (9.8) and Mozambique (9.4) [3]. Among those 11 countries, Ethiopia is one of the best two countries in showing this steady increment. To achieve this

increment, Ethiopia undertook various efforts, including: establishing a health extension program; task shifting and training; community mobilization and engagement; and improved service delivery [2,3,7,9].

In spite of this increment, the overall usage of modern contraception methods in the area is still low relative to most other countries in the world [2,3,7].

Therefore, conducting this study is reasonable to identify the significant determinant for reproductive-aged women's use of modern contraception methods in Ethiopia. Identification of the significant determinant for the usage of modern contraception methods in Ethiopia is useful for reproductive-aged women in deciding whether to pregnant, have a joyful and healthy sexual interaction with averting unwanted pregnancies, the number of their children, and extending of birth interval [10–16]. Reproductive-aged women's use of modern contraception has a great impact on their children and their health. For instance, those reproductive-aged women who don't use modern contraception have a short pregnancy interval, and their children are exposed to the risk of death and undernutrition. Those reproductive-aged women were also exposed highly to miscarriage and pre-eclampsia [13–16].

Evidence shows that the use of modern contraception method was influenced by many of the socio-economic, demographic, health service related, and cultural factors [17–20]. From the study [21], significant variations in the use of modern contraception were observed by wealth status, educational level, visited a health facility, and being informed about family planning. Also, the study [22], shows that women in more urbanized regions such as Addis Ababa, Dire Dawa, and Harari are more likely to use modern contraceptives than respondents in regions that are more rural. The effect of regional variations for religion, place of residence and radio messages further implies that there exists considerable difference in modern contraceptive use among regions and a model with a random coefficient or slope is more appropriate to explain the regional variation than a model with fixed coefficients or without random effects. Also, from another study [23], younger adolescents, married or in a union have low use of modern contraception. From the study [24], maternal age, educational level, wealth index, number of living children, number of births in the last three years, number of under 5 children in the household, religion, and geographic region were independent predictors of modern contraceptive utilization. Some study finding in Nigeria and Egypt reveals that the predictors; place of residence, religion, education, and wealth index have significant effects on the use of modern contraception method [25–29].

Even if many studies have examined the determinant factors associated with utilization of modern contraception methods in Ethiopia through mixed-effect approaches using the 2019 EMDHS data, there are some potential critical gaps in these studies. For instance, the study by:

- [30] Seems limited to the Amhara region, which makes it difficult in offering a more complete picture about all Ethiopians. In other words, we cannot gain an understanding of regional differences and similarities;

- [24] only considers married women, in other words, it leaves out single and unmarried women who might have different circumstances influencing their use of contraceptives; and

- [31,32] does not fully explore the use of multi-level modeling and variable selection.

Hence, to address these limitations, and to further document the significant effect of individual and community level factors on the use of modern contraceptives, this study takes a step further, utilizing a multilevel logistic regression modeling technique. Therefore, the purpose of this study was to determine both individual and community level factors that are associated with the use of modern contraceptives among reproductive-aged women in Ethiopia using a multi-level mixed effects Analysis.

## Methods

### Data and study setting

In this study, we have analyzed data from the 2019 EMDHS (Ethiopian Mini Demographic and Health Survey). Providing up-to-date data on key demographic and health variables was the primary objective of this survey. The data were collected from March 21, 2019, to June 28, 2019, in nine geographical regions and two administrative cities of Ethiopia [1].

### Study design and sampling procedures

Cross Sectional study design were used for EMDHS survey. In the survey, 149,093 EAs (Enumeration Areas) were created to be used as a census frame. In each EAs, on average there are 30 households. A two-stage stratified sample selection strategy was used to take a total of 305 EAs in the first stage and a fixed number of 30 households from each EAs by use of an equal probability systematic selection in the second stage. Then, all women aged 15–49 in each household were interviewed. Finally, the 2019 EMDHS Survey covered 8663 households out of the selected 8,794 households providing a response rate of 99%. About 8,885 women complete the interview from 16,583 women identified for the interview, yielding a response rate of 99% [1]. After a careful multicomponent sexual activity profile (MSAP) of all reproductive-aged women consideration, 8196 reproductive-aged women were identified as sexually active women. Consequently, a total of 8196 eligible reproductive-aged women were included in this study.

### Variables

**Dependent variable.**   The dependent variable (use of modern contraception methods) in this study is a dichotomous variable with category code 1 for those women who use modern contraception and 0 for those who don't use it. In this study, if women use any one of the modern contraception methods such as implants, condoms, injectables, contraceptive pills, standard days method, sterilization, intrauterine devices, emergency contraception, and lactational amenorrhoea, we considered them as a user of modern contraception methods and if not, we considered them as non-users.

**Independent variables.**   The independent variables for contraceptive utilization were based on the previous literature and availability of the variable on 2019 EMDHS dataset. Variables were broadly classified into two main groups, individual level and community level variables aligned for multilevel analytic approach.

**Individual level variables.**   Age of women at the time of survey, education level of women, religion, household wealth index, knowledge of modern contraception methods, marital Status, access to health services, access to media, desire to use modern contraception methods, number of died children, number of living children, family size and total child ever born were included as individual-level variables.

**Community level variables.**   Place of residence and contextual regions were considered at the community level. Contextual variables that were measured at the higher level were considered at the community level. The contextual variables represent the collective social characteristics of the contexts or groups. The contextual region was categorized into City, Pastoralist, and Agrarian and created from nine regions and two city administration with consideration of Addis Abeba, Dire Dawa, and Harar as the City contextual region: Somali and Afar as the Pastoralist contextual region: Tigray, Amhara, Oromia, SNNP, Gambela, and Benishangul Gumuz as the Agrarian contextual region [33].

                                                                          

## Methods of data analysis

**Cross tabulation analysis.** This analysis technique is used in our study to know the percentage of modern contraception users (women) across their socio-economic, demographic, and health characteristics.

**Multilevel logistic regression model.** It is the model in regression analysis with the aim of modeling the relationship between a categorical response variable and any type of independent variable. The dependent variable in this model is dichotomous or binary and the data have hierarchical nature, i.e women was nested in household and household were nested in cluster; So, multilevel binary logistic regression model was done to assess the association between the independent variables and the dependent variable of the study.

In data with a nested structure like that of EMDHS, the individual observations have some degree of correlation within a cluster because of common characteristics they share. As a result, when the correlation with the upper level is ignored and only the individual level characteristics are considered, it might lead to a violation of the assumption of independence between observations [34]. Also, this result will contain biased parameter estimates and will generally lead to underestimation of the standard errors and, produces false significant results and accordingly to incorrect conclusions on effect sizes. To account for the nested nature of EDHS data, a two-level generalized linear mixed model was used. This study had binary outcomes: whether reproductive-aged women used modern contraception or not. We were interested in the probability of modern contraceptive utilization and the influence of individual and regional characteristics. Therefore, to get the mixed effect (fixed effect for both the individual and community level factors and a random effect for the between cluster-variation), a two-level mixed-effect logistic regression analysis was used in this study. Thus, the log of the probability of using modern contraceptives was modeled in the following form:

$$log\left[\frac{\pi_{ij}}{1 - \pi_{ij}}\right] = \beta_0 + \beta_1 X_{ij} + \beta_2 Z_{ij} + U_j$$

Where, $\pi_{ij}$ is the probability of being using modern contraceptives for the $i^{th}$ reproductive-aged women in the $j^{th}$ Cluster. X & Z were individual and community level variables respectively, i & j were the level 1 and level 2 units.

All the fixed effect factors and random effect variables were tested at p-value < 0.025. In the final model, Adjusted Odd Ratio (AHR), P-value, and 95% CI were considered to assess whether each independent variable was statistically significant or not. A variable with P-value ≤ 0.05 was considered as statistically associated with the use of modern contraception methods.

To identify community level effect, Intra Class Correlation (ICC) was valued by the community level variance. Also, the fitness of the model was examined by Proportional Change in Variance (PCV), Median Odds Ratio (MOR) and the Likelihood Ratio Test (LRT).

## Ethics statement

All data that we have used in this study were fully anonymized before we accessed them. Consequently, it is unable to get the respondent in this study. But, permission to use the dataset for this study was got from the Ethiopian Statistics Service after free online registration and then explanation for the use of data. As a result, it doesn't need additional ethical approval. All methods in this study were carried out following relevant guidelines and regulations.

## Results

### Frequency and cross tabulation analysis

From a total of 8196 reproductive-aged women, 2495(30.4%) were using modern contraceptive method and the rest 5701(69.6%) did not use any modern contraceptive methods. Among 2495 contraceptive users, 1657 (67.3%) used injections and 533 (21.7%) used implants/Norplant. The percentage of reproductive-aged women in Ethiopia who were using modern contraception methods was highest for 25–29 aged women (8.2%) than other aged women. The percentage of reproductive-aged women in Ethiopia who used the modern contraception method was higher for those who reside in a rural area (20.2%) than those who reside in an urban area (10.3%). 14.2% of reproductive-aged women in Ethiopia are a follower of the orthodox religion who use the modern contraception method which is highest than other religion followers. In contrast, the use of modern contraception methods is lowest for those reproductive-aged women in Ethiopia who follow traditional religion (0.2%).

The use of modern contraception methods is highest for those reproductive-aged women in Ethiopia at the primary education level (12.9%), while least for women at the highest education level (1.9%).

Reproductive-aged women of Ethiopia in the richest households use modern contraception methods more highly (8.8%) than those women who have fewer income households. The presence of knowledge about modern contraception methods for reproductive-aged women in Ethiopia leads women in using the methods more highly (30.4%) than those who haven't. The desire to use modern contraception methods is also another characteristic of reproductive-aged women on which desirous reproductive-aged women are more highly used than those who haven't desire.

Among reproductive-aged women in Ethiopia, married women used the modern contraception method more highly (28.3%) than women who had another marital status. The result also shows that modern contraception usage is highest for reproductive-aged women who haven't experienced child death (25.3%) than those of have experienced child death. Concerning access to health services, reproductive-aged women who haven't access to health service didn't use modern contraception methods (0.0%).

Reproductive-aged women in Ethiopia used modern contraception methods almost similarly irrespective of access to media.

Those reproductive-aged women in Ethiopia who have below 4 lived children used modern contraception methods more highly (20.4%) than those who had above 4 children (7.1%).

Reproductive-aged women in a 5–8 family-sized household used the modern contraception method more highly (15.7%) than those reproductive-aged women in greater than 9 family-sized households (2.1%).

The result also revealed that 19.2% of reproductive-aged women in Ethiopia are women that have 1–4 children ever born who used modern contraception methods which is highest than other reproductive-aged women who have more than 8 children ever born (1.1%) [Table 1].

### Individual level factors

In model I only individual variables were added. The result signified that age of women, education level, religion, wealth index, number of died children, number of living children, family size and total children ever born had significant association with reproductive-aged women's use of modern contraception method.

**Table 1. Ethiopian reproductive-aged women modern contraception methods usage across their socio- demographic characteristics, EMDHS, 2019.**

| Variables | Category | Usage of modern contraception method | | Total |
|---|---|---|---|---|
| | | Non-Used | Used | |
| | | Count (%) | Count (%) | |
| Age | 15–19 | 1921 (23.4%) | 207(2.5%) | 2128 (26.0%) |
| | 20–24 | 816(10.0%) | 514 (6.3%) | 1330(16.2%) |
| | 25–29 | 769 (9.4%) | 676 (8.2%) | 1445 (17.6%) |
| | 30–34 | 580 (7.1%) | 458 (5.6%) | 1038 (12.7%) |
| | 35–39 | 626 (7.6%) | 363 (4.4%) | 989 (12.1%) |
| | 40–44 | 511 (6.2%) | 199 (2.4%) | 710 (8.7%) |
| | 45–49 | 479 (5.8%) | 77(0.9%) | 556 (6.8%) |
| Place of residence | Urban | 1831 (22.3%) | 842 (10.3%) | 2673 (32.6%) |
| | Rural | 3870 (47.2%) | 1653 (20.2%) | 5523 (67.4%) |
| Education level | No Education | 2287(27.9%) | 992 (12.1%) | 3279(40.0%) |
| | Primary | 2371(28.9%) | 1059(12.9%) | 3430(41.8%) |
| | secondary | 746(9.1%) | 286(3.5%) | 1032(12.6%) |
| | Higher | 299(3.6%) | 157 (1.9%) | 456(5.6%) |
| Religion | Orthodox | 2306(28.1%) | 1166(14.2%) | 3472(42.4%) |
| | Catholic | 25(0.3%) | 21(0.3%) | 46(0.6%) |
| | Protestant | 1478(18.0%) | 761(9.3%) | 2239(27.3%) |
| | Muslim | 1835(22.4%) | 526(6.4%) | 2361(28.8%) |
| | Traditional | 47(0.6%) | 17(0.2%) | 64(0.8%) |
| | Others | 9(0.1%) | 4(0.0%) | 13(0.2%) |
| Wealth index | Poorest | 975(11.9%) | 290(3.5%) | 1265(15.4%) |
| | Poorer | 1076(13.1%) | 415(5.1%) | 1491(18.2%) |
| | Middle | 1014(12.4%) | 530(6.5%) | 1544(18.8%) |
| | Richer | 1227(15.0%) | 539(6.6%) | 1766(21.5%) |
| | Richest | 1410(17.2%) | 721(8.8%) | 2131(26.0%) |
| Knowledge of modern contraception Method | No | 393(4.8%) | 0(0%) | 393(4.8%) |
| | Yes | 5309(64.8%) | 2495(30.4%) | 7804(95.2%) |
| Marital Status | Single | 2272(27.7%) | 48(0.6%) | 2320(28.3%) |
| | Married | 2755(33.6%) | 2320(28.3%) | 5075(61.9%) |
| | Others | 674(8.2%) | 128(1.6%) | 802(9.8%) |
| Current contraceptive method | Not using | 5640(68.8%) | 0(0.0%) | 5640(68.8%) |
| | Injections | 0(0.0%) | 1658(20.2%) | 1658(20.2%) |
| | Implant/Norplant | 0(0.0%) | 533(6.5%) | 533(6.5%) |
| | Others | 61(0.8%) | 304(3.7%) | 365(4.5%) |
| Access to health service | No | 5658(69.0%) | 2(0%) | 5660(69.1%) |
| | Yes | 43(0.5%) | 2493(30.4%) | 2536(30.9%) |
| Access to Media | No | 3491(42.6%) | 1376(16.8%) | 4867(59.4%) |
| | Yes | 2211(27.0%) | 1119(13.7%) | 3330(40.6%) |
| Desire to use modern contraception method | No | 5701(69.6%) | 0(0%) | 5701(69.6%) |
| | Yes | 0(0%) | 2495(30.4%) | 2495(30.4%) |
| Number of died children | Not experience | 4703(57.4%) | 2070(25.3%) | 6773(82.6%) |
| | 1 died | 561(6.8%) | 286(3.5%) | 847(10.3%) |
| | > = 2 died | 437(5.3%) | 139(1.7%) | 576(7.0%) |

*(Continued)*

**Table 1.** (Continued)

| Variables | Category | Usage of modern contraception method | | Total |
|---|---|---|---|---|
| | | Non-Used | Used | |
| | | Count (%) | Count (%) | |
| Number of living children | 0 child | 2673(32.6%) | 242(3.0%) | 2915(35.6%) |
| | 1–4 child | 1799(21.9%) | 1669(20.4%) | 3468(42.3%) |
| | 5–8 child | 1116(13.6%) | 545(6.6%) | 1661(20.3%) |
| | > = 9 child | 114(1.4%) | 39(0.5%) | 153(1.9%) |
| Family size | 1–4 size | 1840(22.5%) | 1035(12.6%) | 2875(35.1%) |
| | 5–8 size | 3104(37.9%) | 1289(15.7%) | 4393(53.6%) |
| | > = 9 size | 757(9.2%) | 170(2.1%) | 927(11.3%) |
| Total child ever born | 0 child | 2644(32.3%) | 230(2.8%) | 2874(35.1%) |
| | 1–4 child | 1619(19.8%) | 1573(19.2%) | 3192(39.0%) |
| | 5–8 child | 1109(13.5%) | 602(7.3%) | 1711(20.9%) |
| | > = 9 child | 328(4.0%) | 90(1.1%) | 418(5.1%) |
| Contextual Region | Pastoralist | 416(5.1%) | 21(0.3%) | 437(5.3%) |
| | Agrarian | 4917(60.0%) | 2347(28.6%) | 7264(88.6%) |
| | City | 368(4.5%) | 126(1.5%) | 494(6.0%) |

## Community level factors

In model II only community level variables were considered. The result signified that residence and contextual region had significant association with reproductive-aged women's use of modern contraception method.

## Full model with individual and community level factors

A two-level mixed-effect logistic regression was used to analyze the effect of women's individual characteristics and community-level factors in determining reproductive-aged women's use of contraception methods. The model comparison result revealed that model III is a better fit for the data as compared to other models, since it has the smallest AIC and BIC values. In this model all individual-level and community-level factors were included.

At a 5% level of significance, the result from this model (two-level binary logistic regression) revealed that the predictors; age of women, education level, religion, wealth index, knowledge of modern contraception method, number of died children, number of living children, family size, total children ever born, residence and contextual region were significantly associated with use of modern contraception method.

There is a 5% decrease in the odds of using modern contraception methods (OR = 0.95, 95% CI 0.94–0.96) among reproductive-aged women in Ethiopia per unit year increase in the age of women by assuming that other predictors remain fixed. Concerning education level, reproductive-aged women in Ethiopia at the primary level of education have 41% more odds of using modern contraception (OR = 1.41, 95% CI 1.21–1.66) than illiterate reproductive-aged women by assuming that other predictors remain fixed. Similarly, reproductive-aged women in Ethiopia at a secondary level of education have 50% more odds of using modern contraception (OR = 1.50, CI 1.19–1.89) than illiterate reproductive-aged women by assuming that other predictors remain fixed. Also, reproductive-aged women in Ethiopia at a above secondary level of education have 113% more odds of using modern contraception (OR = 2.13, CI 1.62–2.80) than illiterate reproductive-aged women by assuming that other predictors remain fixed.

Women's Religion affects use of contraceptive at the individual level. Following Muslim religion for reproductive-aged women in Ethiopia have 47% lower odds of using contraceptive (OR = 0.47, CI 0.39–0.58) than orthodox by assuming that other predictors remain fixed. The other categories of religion were insignificant. Concerning wealth index, poorer reproductive-aged women in Ethiopia have 74% more odds of using modern contraception (OR = 1.74, 95% CI 1.37–2.21) as compared to the poorest reproductive-aged women by assuming that other predictors remain fixed. Economically medium reproductive-aged women in Ethiopia have 117% more odds of using modern contraception (OR = 2.17, 95% CI 1.70–2.80) as compared to the poorest reproductive-aged women by assuming that other predictors remain fixed. Richer reproductive-aged women in Ethiopia have 106% more odds of using modern contraception (OR = 2.06, 95% CI 1.60–2.66) as compared to the poorest reproductive-aged women by assuming that other predictors remain fixed. Richest reproductive-aged women in Ethiopia have 182% more odds of using modern contraception (OR = 2.82, 95% CI 2.14–3.70) as compared to the poorest reproductive-aged women by assuming that other predictors remain fixed.

Child mortality affects use of contraceptive at the individual level. Those reproductive-aged women in Ethiopia who have experiencing 2 or more child death have 29% less odds of using modern contraception (OR = 0.71, 95% CI 0.52–0.98) as compared to women in Ethiopia who haven't experienced child death by assuming that other predictors remain fixed.

Concerning number of living children, the reproductive-aged women who have 1–4 children have 119% more odds of using modern contraception (OR = 2.19, 95% CI 1.07–4.49) than those who haven't a child by assuming that other predictors remain fixed. Those reproductive-aged women in Ethiopia who having 5–8 children have 113% more odds of using modern contraception (OR = 2.13, 95% CI 1.03–4.89) than those who haven't a child by assuming that other predictors remain fixed. Those reproductive-aged women in Ethiopia who having greater than 8 children have 205% more odds of using modern contraception (1.02–9.13) than those who haven't a child by assuming that other predictors remain fixed.

Regards to marital status, married reproductive-aged women in Ethiopia have 29.66 times more odds of using modern contraception (OR = 29.66, 95% CI 20.78, 42.33) than those never in union by assuming that other predictors remain fixed. Those reproductive-aged women in Ethiopia who lived with a partner have 28.73 times more odds of using modern contraception (OR = 28.73, 95% CI 17.04–48.42) than those never in union by assuming that other predictors remain fixed. Divorced reproductive-aged women in Ethiopia have 4.23 times more odds of using modern contraception (OR = 4.23, 95% CI 2.63–6.78) than those never in union by assuming that other predictors remain fixed. Separated reproductive-aged women in Ethiopia have 4.09 times more odds of using modern contraception (OR = 4.09, 95% CI 2.32–7.23) than those never in union by assuming that other predictors remain fixed.

Reproductive-aged women in rural Ethiopia have 17% more odds of using modern contraception (OR = 1.17, 95% CI 1.02–1.59) than reproductive-aged women in urban Ethiopia by assuming that other predictors remain fixed. Contextual Region affects use of contraceptive at the community level. Reproductive-aged women in Agrarian region have 8.05 times more odds of using modern contraception (OR = 8.05, 95% CI 5.57–11.62) as compared to women in pastoralist region. Similarly, reproductive-aged women in City contextual region have 4.35 times more odds of using modern contraception (OR = 4.35, 95% CI 2.87–6.59) as compared to women in pastoralist region by assuming that other predictors remain fixed [Table 2].

## Random effect measures of variation

A two-level mixed-effect logistic regression was used to analyze the effect of women's individual characteristics and community-level factors in determining women's use of contraceptives.

**Table 2. Two-level logistic regression output for predictor of modern contraceptive use among reproductive-aged women in Ethiopia, 2019 EMDHS.**

| Variables | Model 1 AOR (95%CI) | Model 2 AOR (95% CI) | Model 3 AOR (95% CI) | Model 4 AOR (95% CI) | P-value for AOR in Model 4 |
|---|---|---|---|---|---|
| **Individual Level factors** | | | | | |
| Age | | 0.94(0.93–0.95) | | 0.95(0.94–0.96) | <0.001 |
| **Education Level** | | | | | |
| No education | | 1 | | 1 | |
| Primary | | 1.36(1.16–1.60) | | 1.41(1.21–1.66) | <0.001 |
| Secondary | | 1.47(1.17–1.85) | | 1.50(1.19–1.89) | 0.001 |
| Higher | | 2.13(1.62–2.80) | | 2.13(1.62–2.80) | <0.001 |
| **Religion** | | | | | |
| Orthodox | | 1 | | 1 | |
| Catholic | | 0.92(0.47–1.78) | | 0.95(0.49–1.84) | 0.874 |
| Protestant | | 0.82(0.66–1.02) | | 0.87(0.70–1.09) | 0.228 |
| Muslim | | 0.64(0.52–0.78) | | 0.47(0.39–0.58) | <0.001 |
| Traditional | | 0.48(0.21–1.13) | | 0.54(0.23–1.28) | 0.161 |
| Other | | 1.04(0.37–2.93) | | 1.13(0.40–3.22) | 0.817 |
| **Wealth index** | | | | | |
| Poorest | | 1 | | 1 | |
| Poorer | | 1.43(1.12–1.81) | | 1.74(1.37–2.21) | <0.001 |
| Middle | | 1.79(1.40–2.29) | | 2.17(1.70–2.80) | <0.001 |
| Richer | | 1.74(1.35–2.26) | | 2.06(1.60–2.66) | <0.001 |
| Richest | | 2.82(2.04–3.91) | | 2.82(2.14–3.70) | <0.001 |
| **Child mortality** | | | | | |
| Not experience | | 1 | | 1 | |
| One died | | 0.86(0.70–1.06) | | 0.86(0.70–1.07) | 0.171 |
| ≥ 2 died | | 0.71(0.51–0.97) | | 0.71(0.52–0.98) | 0.040 |
| **Living children** | | | | | |
| No child | | 1 | | 1 | |
| 1–4 child | | 2.12(1.03–4.36) | | 2.19(1.07–4.49) | 0.033 |
| 5–8 child | | 2.06(0.90–4.75) | | 2.13(0.93–4.89) | 0.075 |
| ≥ 9 | | 2.94(0.98–8.80) | | 3.05(1.02–9.13) | 0.046 |
| **Family size** | | | | | |
| 1–4 size | | 1 | | 1 | |
| 5–8 size | | 0.87(0.76–0.99) | | 0.89(0.77–1.02) | 0.093 |
| ≥ 9 size | | 0.67(0.52–0.86) | | 0.68(0.53–0.88) | 0.003 |
| **Child ever born** | | | | | |
| 0 child | | 1 | | 1 | |
| 1–4 child | | 10.69(5.10–22.42) | | 10.51(5.02–21.98) | <0.001 |
| 5–8 child | | 13.72(5.77–32.62) | | 13.65(5.75–32.41) | <0.001 |
| ≥ 9 child | | 14.39(5.06–40.93) | | 13.98(4.92–39.70) | <0.001 |
| **Marital status** | | | | | |
| Single | | 1 | | 1 | |
| Married | | 39.51(29.7–52.7) | | 34.65(24.3–49.35) | <0.001 |
| Other | | 7.07(5.03–9.93) | | 6.53(4.41–9.67) | 0.002 |
| **Community level factors** | | | | | |
| **Place of residence** | | | | | |
| Urban | | | 1 | 1 | |
| Rural | | | 0.69(0.52–0.90) | 1.07(0.76–1.49) | 0.711 |

*(Continued)*

**Table 2.** (Continued)

| Variables | Model 1 AOR (95%CI) | Model 2 AOR (95% CI) | Model 3 AOR (95% CI) | Model 4 AOR (95% CI) | P-value for AOR in Model 4 |
|---|---|---|---|---|---|
| **Contextual region** | | | | | |
| Pastoralist | | | 1 | 1 | |
| Agrarian | | | 8.05(5.6–11.6) | 6.02(3.92–9.26) | <0.001 |
| City | | | 4.35(2.87–6.59) | 3.37(2.11–5.35) | <0.001 |
| Constant | 0.28(0.25–0.32) | | | | |

The level of significance that we used: 5%.

The ICC (%) in the empty model indicates that 21% of the total variance in the odds of contraceptive use was accounted by between cluster variations of characteristics. The between cluster variability declined over successive models from 21% in the empty model into 18% in individual-level only model, 15% in community-level factors only model and 14% in the combined model. Also as we have seen the variance of a constant was declined over successive models from 0.89 in null model, 0.72 in individual level only model, 0.59 in community-level factors only and 0.55 in the full model [Table 3]. The AIC & BIC values were also small for the full model as compared to the other models. Thus, the combined model of individual-level and community-level factors was preferred for predicting reproductive-aged women's contraceptive use.

## Discussion

Contraceptives prevent unwanted and unplanned pregnancies, as well as complications, arising from these conditions. The intricacies surrounding contraceptive decisions are revealed by the detailed examination of determinant factors influencing the use of modern contraceptives among women of reproductive age. Using multi-level modeling techniques, especially the stepwise approach, helps to identify patterns and outlines the complex interactions between different predictors.

The model comparison results revealed that model III (random intercept random slop model) was a better fit for the data as compared to other models, since it has the smallest AIC and BIC values. This is consistent with other studies' findings [24,34–36] that highlight the effectiveness of multilevel logistic regression as a more powerful analytical tool than traditional logistic regression analyses for analyzing hierarchical data structures.

Many studies were done about the use of modern contraception methods among women in most populated countries in Africa like Nigeria, Ethiopia and Egypt. Some study finding in

**Table 3. Measure of variation on individual and community level predictors among reproductive-aged women in Ethiopia, EMDHS-2019.**

| Measure of variation | Model 1(Null model) | Model 2 | Model 3 | Model 4 |
|---|---|---|---|---|
| Variance (SE) | 0.89 | 0.72 | 0.55 | 0.59 |
| ICC (%) | 0.21 | 0.18 | 0.14 | 0.15 |
| **Model fit statistics** | | | | |
| Log-likelihood | -4340.33 | -3628.77 | -4272.73 | -3589.34 |
| AIC | 8684.67 | 7307.55 | 8555.46 | 7234.67 |
| BIC | 8698.66 | 7482.54 | 8590.46 | 7430.66 |

The level of significance that we used: 5%.

Nigeria and Egypt reveals that the predictors; place of residence, religion, education, and wealth index have significant effects on the use of modern contraception method [21,25–29]. These results coincide with our study findings. The robustness and consistency of these predictors across various research endeavors are demonstrated by these coinciding results. This increases trust in their significance for comprehending and addressing the use of contraception in Africa. In addition to these factors, in our study, age and number of living children are significant factors which both are not significant in all aforementioned studies. Nonetheless, the disparity in results reveals the intricacy present in these demographic variables among various geographical areas and age groups. This difference may be the result of educational differences between age groups in these countries, emphasizing the importance of education as a key factor.

Furthermore, when compared to the 69.6% utilization rate found in the current study, the stark differences in the percentages of modern contraceptive use among various sub-Saharan countries—Angola (60.7%), Kenya (34.5%), Uganda (26.6%), and Burundi (29.78%)—indicate complex cultural, traditional, and normative influences shaping contraceptive practices [37–40]. In addition to their years of study difference, these variations most likely result from different historical settings, cultural norms, and educational levels among reproductive-age women in these areas.

In this study reproductive-aged women in City contextual region have more odds of using modern contraception as compared to women in pastoralist region. This result is consistent with another study [22]. Their finding shows that women in more urbanized regions such as Addis Ababa, Dire Dawa, and Harari are more likely to use modern contraceptives than respondents in regions that are more rural. The effect of regional variations for religion, place of residence and radio messages further implies that there exists considerable deference in modern contraceptive use among regions and a model with a random coefficient or slope is more appropriate to explain the regional variation than a model with fixed coefficients or without random effects. In this study younger women have used more contraception as compared to older. This finding was contradicted with another study [23]. In their finding younger adolescents, married or in a union have low use of modern contraception.

## Strength and weakness of the study

To overcome the problem of using an ordinary model for EMDS data (nested structure in nature), we used an advanced model (two-level generalized linear mixed). Moreover, the result from this model provide unbiased parameter estimates, precise standard error, true significant result and correct conclusions on effect sizes even if we have used data with hierarchical nature.

Since, EMDS is a mini survey; some important predictors for the use of modern contraception were not included in this study. This may have an effect on the parameter estimate value of the model and on their significant effect.

## Conclusion

In this study, the main objective was to identify the significant determinant factors in the use of modern contraception methods among reproductive-aged women in Ethiopia. From a total of 8196 reproductive-aged women included in the study, 2495(30.4%) were using modern contraceptive method and the rest 5701(69.6%) did not use any modern contraceptive methods. Reproductive-aged women in Ethiopia with more living children, residing in urban/agrarian region, younger, wealthier, married, and more educated, were more likely to be modern contraceptive users. The concerned bodies in Ethiopia should bring forward the intervention

strategy and should expand the existed programs to improve the use of modern contraception methods among non-pregnant reproductive-aged women in Ethiopia. Especially, they should give special attention to non-pregnant reproductive-aged women of less income, resident in pastoralist region, less educated, unmarried, and haven't living child.

## Author Contributions

**Conceptualization:** Molalign Gualu Gobena.

**Data curation:** Molalign Gualu Gobena, Maru Zewdu Kassie.

**Formal analysis:** Molalign Gualu Gobena, Maru Zewdu Kassie.

**Investigation:** Molalign Gualu Gobena, Maru Zewdu Kassie.

**Methodology:** Molalign Gualu Gobena, Maru Zewdu Kassie.

**Software:** Molalign Gualu Gobena.

**Validation:** Molalign Gualu Gobena, Maru Zewdu Kassie.

**Writing – original draft:** Molalign Gualu Gobena, Maru Zewdu Kassie.

**Writing – review & editing:** Molalign Gualu Gobena, Maru Zewdu Kassie.

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
