## [Decision Letter · Decision Letter 0]

6 Dec 2023

PONE-D-23-29235Determinants of the use of modern contraceptives among non-pregnant reproductive-aged women in Ethiopia: A multi-level mixed effects AnalysisPLOS ONE

Dear Dr. Gobena,

Thank you for submitting your manuscript to PLOS ONE. After careful consideration, we feel that it has merit but does not fully meet PLOS ONE’s publication criteria as it currently stands. Therefore, we invite you to submit a revised version of the manuscript that addresses the points raised during the review process.

We look forward to receiving your revised manuscript.

Kind regards,

Gizachew Gobebo Mekebo

Academic Editor

PLOS ONE

Journal Requirements:

Reviewers' comments:

Reviewer's Responses to Questions

**Comments to the Author**

1. Is the manuscript technically sound, and do the data support the conclusions?

Reviewer #1: Yes

Reviewer #2: No

2. Has the statistical analysis been performed appropriately and rigorously? 

Reviewer #1: Yes

Reviewer #2: Yes

3. Have the authors made all data underlying the findings in their manuscript fully available?

Reviewer #1: No

Reviewer #2: No

4. Is the manuscript presented in an intelligible fashion and written in standard English?

Reviewer #1: No

Reviewer #2: No

5. Review Comments to the Author

Reviewer #1: Reviewer Comments to authors:

The authors of the paper, " Determinants of the use of modern contraceptives among non-pregnant reproductive-aged women in Ethiopia: A multi-level mixed effects Analysis“, structured the paper nicely covering both statistical theories on a multi-level mixed effects Analysis and its application to the use of modern contraceptives among non-pregnant reproductive-aged women in Ethiopia. I found the paper contributes little to either statistical modeling or the use of modern contraceptives literature. I found some issues discussed below:

1. In the title: Why did you say non-pregnant? It’s better to update the title like, “Determinants of the use of modern contraceptives among women of reproductive age group in Ethiopia: A multi-level mixed effects Analysis”. Because, it’s obvious that pregnant women do not use contraceptive methods.

2. In the Abstract: The authors applied a multi-level mixed effects Analysis to identify determinants of the use of modern contraceptives by simply mentioning multi-level mixed effects analysis without any appropriate reason. Multi-level modeling can be understood by a statistician but not the researchers with non-statistics background. Since the research does not focus on any statistical contribution rather then their application, the authors should mention at the beginning of the paper what and why multilevel modeling is necessary for such a study.

3. In the Introduction: First of all, the introduction lacks literature review on multi-level mixed effects analysis compared to the single level logistic regression analysis. I wonder the authors write "Even if several studies had examined and documented about the determinant factors associated with the use of modern contraception methods in Ethiopia, the impact of community level factors on it had received less consideration." Because, many studies have been done on the use of modern contraceptive among women of reproductive age group in Ethiopia using multilevel mixed effects analysis and 2019 EMDHS (The 2019 Ethiopian Mini Demographic and Health Survey) data. I don't know the importance of this study (your study). So, please state the gaps clearly authors have not stated clearly the gaps with many previous related studies in Ethiopia. Please state the holes clearly in this section.

- On page 2, lines 31& 32; you have said, “Among those 11 countries, Ethiopia is the best two countries in showing this steady increment.” What do you mean? Please correct it by, “Among those 11 countries, Ethiopia is one of the best two countries in showing this steady increment.”

- On page 2, line 52; please correct the word “deference” by “difference”. In general, please enhance the quality of your paper by improving the English language (grammar, spelling and coherence of the sentences or paragraphs).

4. In the results:

- Please specify the level of significance at the beneath of table 2 and table 3.

- Incorporate statistical findings, such as charts and maps, into your content.

Reviewer #2: Dear Editor, Thank you very much for your warm invitation to this interesting authors work. I appreciate the authors of all their efforts. The study seems rigorous but it has many issues must be addressed before publication. Find my comments and suggestion hereunder.

Title

What is the importance of pronouncing “non-pregnant” ?

Introduction

The authors mentioned that lactational amenorrhoea is a modern contraceptive method. Lactational amenorrhea method natural/or traditional method rather than modern method.

The authors did not mention all the aims of family planning utilization.

This section lacks consistency of ideas

The authors did not describe the current figures of modern contraceptive utilization in Ethiopia.

What efforts were made to boost the uptake of the modern contraceptive methods in the area?

Line 126-131. You narrated general science. It would be better if you focused on your data analysis process.

Results

Table 2 doesn’t clearly show statistically significant associations because it has no p-value.

Discussion

This section is too shallow.

Major comment

What makes your findings different from the 2019 Mini EDHS report?

The authors need to address grammar and typo errors

It seems authors focused on statistical analysis sections. The background and discussion sections should be written and interpreted by experts in the area.

6. PLOS authors have the option to publish the peer review history of their article (what does this mean?). If published, this will include your full peer review and any attached files.

Reviewer #1: No

Reviewer #2: No

---

## [Author Response · Author response to Decision Letter 0]

30 Dec 2023

Point by point author’s response to comments and questions

Reviewer #1 comments/questions with corresponding author’s response

1) In the title: Why did you say non-pregnant? It’s better to update the title like, “Determinants of the use of modern contraceptives among women of reproductive age group in Ethiopia: A multi-level mixed effects Analysis”. Because, it’s obvious that pregnant women do not use contraceptive methods.

Author’s response: Thank you! We have appreciated your suggestion and modified the title based on your comment.

2) In the Abstract: The authors applied a multi-level mixed effects Analysis to identify determinants of the use of modern contraceptives by simply mentioning multi-level mixed effects analysis without any appropriate reason. Multi-level modeling can be understood by a statistician but not the researchers with non-statistics background. Since the research does not focus on any statistical contribution rather then their application, the authors should mention at the beginning of the paper what and why multilevel modeling is necessary for such a study.

Author’s response: Thank you for your comment. 

Abstract:

You are entirely correct. We recognize that the abstract needs to provide a clearer justification for the use of multilevel mixed-effects analysis (MLMEA). We have updated it to give a succinct justification of why multi-level modeling was selected as the best model for this investigation, emphasizing its benefit in taking community-level influences on individual contraceptive use into account. With this addition, the abstract will be more readable by researchers who are not statistical experts.

3) In the Introduction: First of all, the introduction lacks literature review on multi-level mixed effects analysis compared to the single level logistic regression analysis. I wonder the authors write "Even if several studies had examined and documented about the determinant factors associated with the use of modern contraception methods in Ethiopia, the impact of community level factors on it had received less consideration." Because, many studies have been done on the use of modern contraceptive among women of reproductive age group in Ethiopia using multilevel mixed effects analysis and 2019 EMDHS (The 2019 Ethiopian Mini Demographic and Health Survey) data. I don't know the importance of this study (your study). So, please state the gaps clearly authors have not stated clearly the gaps with many previous related studies in Ethiopia. Please state the holes clearly in this section.

On page 2, lines 31& 32; you have said, “Among those 11 countries, Ethiopia is the best two countries in showing this steady increment.” What do you mean? Please correct it by, “Among those 11 countries, Ethiopia is one of the best two countries in showing this steady increment.”

On page 2, line 52; please correct the word “deference” by “difference”. In general, please enhance the quality of your paper by improving the English language (grammar, spelling and coherence of the sentences or paragraphs). 

Author’s response: Thank you for your critical review and constructive comment! 

Introduction:

Literature Review on MLMEA: We concur that it will be helpful to include a succinct review of the literature on MLMEA. We've included a special paragraph comparing and explaining why, because of its capacity to take into account both individual and community-level factors, MLMEA were chosen for our study.

Gaps in Previous Research: We apologize for the unclear articulation of research gaps. We have revised the introduction to clearly identify specific limitations in existing studies on modern contraceptive use in Ethiopia, particularly emphasizing the lack of focus on community-level influences. We have also clarified how our study addresses these gaps by employing MLMEA and analyzing relevant community-level variables.

Sentences correction: we have corrected the sentence based on your comment.

Word correction: We have rectified the typo "deference" to "difference". 

We have carefully proofread the whole document and made every effort to improve the English language throughout, as advised.

4) In the results:

-Please specify the level of significance at the beneath of table 2 and table 3.

-Incorporate statistical findings, such as charts and maps, into your content.

Author’s response: This is also another constructive comment. 

Results:

Statistical Significance: We have added a footnote showing the level of significance below Tables 2 and 3 to facilitate the interpretation of the results. 

Visualization: We understand how important it is to include visual components like maps and charts. But, instead of it, we have used cross-tabulation to show descriptive statistics. 

Reviewer #2 comments/questions with corresponding author’s response 

Title 

1) What is the importance of pronouncing “non-pregnant”? 

Author’s response: Thank you for question! We have understood your question and changed the original title to "Determinants of the use of modern contraceptives among women of reproductive age group in Ethiopia: A multi-level mixed effects analysis".

2) Introduction

The authors mentioned that lactational amenorrhoea is a modern contraceptive method. Lactational amenorrhea method natural/or traditional method rather than modern method.

The authors did not mention all the aims of family planning utilization.

This section lacks consistency of ideas

The authors did not describe the current figures of modern contraceptive utilization in Ethiopia. 

What efforts were made to boost the uptake of the modern contraceptive methods in the area?

Line 126-131. You narrated general science. It would be better if you focused on your data analysis process. 

Author’s response: Thank you for your constructive comment!

Introduction:

Lactational Amenorrhea: You are entirely correct. We regret that lactational amenorrhea was incorrectly categorized as a modern technique. As recommended, we have revised it. 

Aims of Family Planning: We have recognized your point and expanded the introduction by including all key aims of contraceptive utilization. 

Consistency of Ideas: We made an effort to improve the introduction in order to guarantee a coherent flow of concepts and remove abrupt changes. Moreover, we have logically grouped related information to improve the overall coherence.

 Modern Contraceptive Utilization Figures: Please note that we have tried to provide the most up-to-date information (i.e. before 2021). Beyond this year the figures for modern contraceptive utilization in Ethiopia haven't been published.

Boosting Contraceptive Uptake: We have addressed your question by including relevant information. 

Data Analysis Process (Cross-Tabulation): We have revised the explanation under cross-tabulation section (in lines 126-131) by focusing on our data analysis process.

3) Results

Table 2 doesn’t clearly show statistically significant associations because it has no p-value.

Author’s response: Thank you! 

Results:

Table 2: We have appreciated your suggestion and added a column displaying p-values in the final fitted model (model 4) alongside confidence intervals. Please note that putting p-value for estimates in all models (model 1-4) consumes space. Instead of it, we have provided a confidence interval for each estimate in all models. It helps to identify statistically significant associations and give more comprehensive information than p-value.

4) Discussion 

This section is too shallow. 

Author’s response: Thank you for your comment and suggestion. We have tried to make the discussion section to be deeper and more insightful. 

5) Major comment 

What makes your findings different from the 2019 Mini EDHS report? 

The authors need to address grammar and typo errors 

It seems authors focused on statistical analysis sections. The background and discussion sections should be written and interpreted by experts in the area.

Author’s response: Thank you again for your major comment. 

Difference from the 2019 Mini EDHS report:

The findings from this study differed in many ways from 2019 Mini EDHS report. Some of them can be expressed as follows:

Findings from this study 

1) Based on Multi-level analysis

2) Provide both descriptive and inferential statistics 

3) Focus on reproductive aged women within Ethiopia

4) Have a policy implication by targeting this specific sub-population within Ethiopia

5) Provide the effect at community and individual level

Findings from 2019 Mini EDHS report

1) Potentially based on single-level analysis

2) Provide only descriptive statistics 

3) Focus on the whole population within Ethiopia

4) Have a policy implication by targeting the whole population within Ethiopia

5) Provide the effect only at individual level

Grammar and Typos: After carefully going over the entire manuscript, we made an effort to ensure that there were no typos or grammatical errors. 

Expertise in Background and Discussion: We fully agree that expertise in the relevant field is crucial for these sections. But, please notice that we ourselves are both statistician and public health expert.

---

## [Decision Letter · Decision Letter 1]

8 May 2024

PONE-D-23-29235R1Determinants of the use of modern contraceptives among women of reproductive age group in Ethiopia: A multi-level mixed effects AnalysisPLOS ONE

Dear Dr. Gobena,

Thank you for submitting your manuscript to PLOS ONE. After careful consideration, we feel that it has merit but does not fully meet PLOS ONE’s publication criteria as it currently stands. Therefore, we invite you to submit a revised version of the manuscript that addresses the points raised during the review process.

We look forward to receiving your revised manuscript.

Kind regards,

Kahsu Gebrekidan

Academic Editor

PLOS ONE

Reviewers' comments:

Reviewer's Responses to Questions

**Comments to the Author**

1. If the authors have adequately addressed your comments raised in a previous round of review and you feel that this manuscript is now acceptable for publication, you may indicate that here to bypass the “Comments to the Author” section, enter your conflict of interest statement in the “Confidential to Editor” section, and submit your "Accept" recommendation.

Reviewer #1: All comments have been addressed

Reviewer #2: (No Response)

2. Is the manuscript technically sound, and do the data support the conclusions?

Reviewer #1: Yes

Reviewer #2: No

3. Has the statistical analysis been performed appropriately and rigorously? 

Reviewer #1: No

Reviewer #2: I Don't Know

4. Have the authors made all data underlying the findings in their manuscript fully available?

Reviewer #1: Yes

Reviewer #2: Yes

5. Is the manuscript presented in an intelligible fashion and written in standard English?

Reviewer #1: Yes

Reviewer #2: No

6. Review Comments to the Author

Reviewer #1: The authors tried to address all the questions raised. But they have not included any graphs or charts in the result part.

Reviewer #2: Recommendation for authors

Authors utilized the 2019 mini EDHS data of modern contraceptive methods. The proportion of modern contraceptive method utilization according to the mini EDHS was 41%. However, the authors found 30.4%. The same is true for each method. How this could be?

Moreover, how did the authors found those all individual and community levels variables? The 2019 mini EDHS data is limited a certain variables.

Title: it would be “Determinants of modern contraceptives utilization among reproductive age women in Ethiopia: A multi-level mixed effects Analysis”

Abstract: The authors concluded that young, wealthy, married, educated, fertile and, residents of City and Agrarian region reproductive-aged women in Ethiopia were relatively the highest users of modern contraception methods. I wonder how they reached to this finding. Do infertile women use any contraceptive method practically? Again the EDHS only include married women study, if so, how did you compare it with unmarried women?

Discussion

Line 323: Avoid etc. or mention all the rest findings.

Although authors found various significant variables, they did not discuss them all briefly.

7. PLOS authors have the option to publish the peer review history of their article (what does this mean?). If published, this will include your full peer review and any attached files.

Reviewer #1: No

Reviewer #2: No

---

## [Author Response · Author response to Decision Letter 1]

18 May 2024

To: Editor-in-Chief/Academic Editor

PLOS ONE Journal 

Dear Editor-in-Chief /Academic Editor,

Greetings!

Re: Manuscript Submission - [Determinants of the use of modern contraceptives among women of reproductive age group in Ethiopia: A multi-level mixed effects analysis] 

We thank the reviewers and editors for their insightful feedback and recommendations on our manuscript, "Determinants of the use of modern contraceptives among women of reproductive age group in Ethiopia: A multi-level mixed effects analysis". We value the time and work you have taken to evaluate our work for the second time. We have addressed all of their points with revisions in response to their comments. We address every point raised by the reviewers in the section that follows.

We think that these changes have strengthened our manuscript considerably and taken care of all the reviewers' concerns. We hope that our updated work satisfies PLOS ONE Journal standards and is ready for publication. Thank you for taking a look at our updated manuscript. 

Thank you once again for your time and attention to our submission.

Sincerely,

Molalign Gualu Gobena,

Principal/Corresponding Author

Point by point author’s response to comments and questions

Reviewer #1 comments/questions with corresponding author’s response

General Comments

The authors tried to address all the questions raised. But they have not included any graphs or charts in the result part.

Author’s response: Thank you for your continued review and feedback. We appreciate your suggestion to include graphs or charts in the results section.

We admit that charts and graphs are effective tools for presenting data because they are visual. But in this instance, we chose to employ cross-tabulations for the reasons listed below:

 Number of Variables: Our study involved a relatively large number of socio-demographic factors potentially influencing modern contraceptive use. Presenting all these variables in separate graphs could be visually overwhelming and potentially obscure key findings. In addition, if we used graphs or charts in this manuscript, the number of figures becomes more than the number of figures allowed by the guideline of the journal.

Detailed Information: Cross-tabulations allow us to present detailed breakdowns of contraceptive use by each variable category (e.g., Age, education level, religion, household wealth index, etc…). This level of detail is crucial for understanding the specific relationships between these factors and contraceptive use.

Clarity for Complex Relationships: While graphs are excellent for showcasing trends, cross-tabulations are often better suited for illustrating complex interactions between multiple variables. This can be particularly relevant when analyzing data on social issues like contraceptive use.

Reviewer #2 comments/questions with corresponding author’s response 

1) In the Title; 

Title: it would be “Determinants of modern contraceptives utilization among reproductive age women in Ethiopia: A multi-level mixed effects Analysis”

Author’s response: I appreciate the title suggestion you made. We value your suggestions on how to make it more succinct and clear. The title "Determinants of modern contraceptives utilization among reproductive age women in Ethiopia: A multi-level mixed effects analysis" is the one that the reviewer preferred. We can understand that. We respectfully suggest retaining the original title, "Determinants of the use of modern contraceptives among women of reproductive age group in Ethiopia: A multi-level mixed effects analysis," even though it appropriately captures the essence of the study.

Here's our reasoning behind this preference:

Clarity: Our original title explicitly mentions "use" of modern contraceptives, which some readers might find easier to understand compared to "utilization."

Target Audience: Since the word "use" is frequently used in this context in Ethiopia, we think the original wording may be more understandable to policymakers and medical professionals there.

Consistency: The terminology used in the original title and the rest of the manuscript are in good alignment.

2) In the Abstract;

Abstract: The authors concluded that young, wealthy, married, educated, fertile and, residents of City and Agrarian region reproductive-aged women in Ethiopia were relatively the highest users of modern contraception methods. I wonder how they reached to this finding. Do infertile women use any contraceptive method practically? Again the EDHS only include married women study, if so, how did you compare it with unmarried women?

Author’s response: No. We apologize the confusion we made in the word “fertile” of this statement. We were used this word to mean women with more living children by considering the variable “living children”. We have revised it. In our study there is no a variable category name “fertile women” compared to infertile women. Please look the categories of the variable “living children” in the table 2. In this variable, the reference category is “No Child”. This is the reason that we were already tried to conclude by referring this reference category. Similarly, married women were not also compared with unmarried women. Rather it is compared with single women. This is because as it is a reference category of the variable “Marital status”. Please look it also from this table. 

The EDHS does not include only married women. For detail, please look it on my response at result section. 

3) In the Methods;

Moreover, how did the authors found those all individual and community levels variables? 

Author’s response: We have considered variables measured at individual level as individual level variables whereas contextual variables that were measured at the higher level as a community level variables.

The 2019 mini EDHS data is limited a certain variables. 

Author’s response: Yes, but, we have used the available variable which are of our variables of interest in the study.

4) In the Results;

Authors utilized the 2019 mini EDHS data of modern contraceptive methods. The proportion of modern contraceptive method utilization according to the mini EDHS was 41%. However, the authors found 30.4%. The same is true for each method. How this could be?

Author’s response: Please note that the 2019 Mini EDHS does not only include married women. It includes all reproductive aged women (15-49). Please look it again the report on page 37 under the title “5.1 CONTRACEPTIVE KNOWLEDGE AND USE”. Please give attention the two shaded sentences under this title. Hence, the findings under this title make use of “All women age 15-49 and currently married women aged 15-49” as a sample. I think your comment on percentage of modern method is based on the report sentence “Many currently married women use a modern method (41%), while only 1% use a traditional method” on page 38. This statistics is not applied for all reproductive aged women. This is only applied only for currently married women. To find the full finding, please look the Table 5.3 on page 43. In this table the percentage of modern contraceptive use among all reproductive aged women are 28.1%. But, in our study it is 30.4%. The possible differences may be we used only 8196 reproductive aged women where as they used 8885 reproductive aged women as a sample. This implies we remove 689 cases in our analysis. This is because we were made a careful multicomponent sexual activity profile (MSAP) of all reproductive-aged women consideration to identify sexually active women. As a result, we have identified 8196 reproductive-aged women as sexually active women. 

5) In the Discussion;

Line 323: Avoid etc. or mention all the rest findings.

Although authors found various significant variables, they did not discuss them all briefly.

Author’s response: Thank you for your continued review and feedback. We appreciate you highlighting the importance of comprehensive discussion in the manuscript. In our first revision, we felt as we have discussed all significant variables found in the study, albeit briefly. We understand that conciseness is important, but we also want to ensure all relevant findings are addressed. In line 323, we have avoided etc.

---

## [Decision Letter · Decision Letter 2]

21 Jun 2024

Determinants of the use of modern contraceptives among women of reproductive age groupin Ethiopia: A multi-level mixed effects Analysis

PONE-D-23-29235R2

Dear Mr. Molalign,

We’re pleased to inform you that your manuscript has been judged scientifically suitable for publication and will be formally accepted for publication once it meets all outstanding technical requirements.

Kind regards,

Kahsu Gebrekidan, Ph.D.

Academic Editor

PLOS ONE

Additional Editor Comments (optional):

Reviewers' comments:

Reviewer's Responses to Questions

**Comments to the Author**

1. If the authors have adequately addressed your comments raised in a previous round of review and you feel that this manuscript is now acceptable for publication, you may indicate that here to bypass the “Comments to the Author” section, enter your conflict of interest statement in the “Confidential to Editor” section, and submit your "Accept" recommendation.

Reviewer #2: All comments have been addressed

2. Is the manuscript technically sound, and do the data support the conclusions?

Reviewer #2: Yes

3. Has the statistical analysis been performed appropriately and rigorously? 

Reviewer #2: Yes

4. Have the authors made all data underlying the findings in their manuscript fully available?

Reviewer #2: Yes

5. Is the manuscript presented in an intelligible fashion and written in standard English?

Reviewer #2: Yes

6. Review Comments to the Author

Reviewer #2: Authors addressed all comments. So the manuscript is suitable for publication in the current status.

7. PLOS authors have the option to publish the peer review history of their article (what does this mean?). If published, this will include your full peer review and any attached files.

Reviewer #2: No
